# Compaction and segregation of sister chromatids via active loop extrusion

Anton Goloborodko[1], Maxim V Imakaev[1], John F Marko[2,4], Leonid Mirny[1,3]*

[1]Department of Physics, Massachusetts Institute of Technology, Cambridge, United States; [2]Department of Molecular Biosciences, Northwestern University, Evanston, United States; [3]Institute for Medical Engineering and Science, Massachusetts Institute of Technology, Cambridge, United States; [4]Department of Physics and Astronomy, Northwestern University, Evanston, United States

**Abstract** The mechanism by which chromatids and chromosomes are segregated during mitosis and meiosis is a major puzzle of biology and biophysics. Using polymer simulations of chromosome dynamics, we show that a single mechanism of loop extrusion by condensins can robustly compact, segregate and disentangle chromosomes, arriving at individualized chromatids with morphology observed in vivo. Our model resolves the paradox of topological simplification concomitant with chromosome 'condensation', and explains how enzymes a few nanometers in size are able to control chromosome geometry and topology at micron length scales. We suggest that loop extrusion is a universal mechanism of genome folding that mediates functional interactions during interphase and compacts chromosomes during mitosis.

*For correspondence: leonid@
mit.edu

**Competing interests:** The
authors declare that no
competing interests exist.

**Reviewing editor:** Antoine M
van Oijen, University of
Groningen, Netherlands

## Introduction

The mechanism whereby eukaryote chromosomes are compacted and concomitantly segregated from one another remains poorly understood. A number of aspects of this process are remarkable. First, the chromosomes are condensed into elongated structures that maintain the linear order, i.e. the order of genomic elements in the elongated chromosome resembles their order along the genome (*Trask et al., 1993*). Second, the compaction machinery is able to distinguish different chromosomes and chromatids, preferentially forming intra-chromatid cross-links: if this were not the case, segregation would not occur (*Nasmyth, 2001*). Third, the process of compaction is coincident with segregation of sister chromatids, i.e. formation of two separate chromosomal bodies. Finally, originally intertwined sister chromatids become topologically disentangled, which is surprising, given the general tendency of polymers to become *more* intertwined as they are concentrated (*Marko, 2011*).

None of these features cannot be produced by indiscriminate cross-linking of chromosomes (*Marko and Siggia, 1997*), which suggests that a novel mechanism of polymer compaction must occur, namely 'lengthwise compaction' (*Marko, 2009*; *2011*; *Marko and Rippe, 2011*), which permits each chromatid to be compacted while avoiding sticking of separate chromatids together. Cell-biological studies suggest that topoisomerase II and condensin are essential for metaphase chromosome compaction (*Hirano and Mitchison, 1993*; *1994*; *Wood and Earnshaw, 1990*; *Hirano, 1995*), leading to the hypothesis that mitotic compaction-segregation relies on the interplay between the activities of these two protein complexes. Final structures of mitotic chromosomes were shown to consist arrays of consecutive loops (*Paulson and Laemmli, 1977*; *Earnshaw and Laemmli, 1983*; *Naumova et al., 2013*). Formation of such arrays would naturally results in lengthwise chromosome compaction.

One hypothesis of how condensins can generate compaction without crosslinking of topologically distinct chromosomes is that they bind to two nearby points and then slide to generate a progressively larger loop (*Nasmyth, 2001*). This 'loop extrusion' process creates an array of consecutive loops in individual chromosomes (*Nasmyth, 2001*). When loop-extruding condensins exchange with the solvent, the process eventually settles at a dynamical steady state with a well-defined average loop size (*Gerlich et al., 2006*; *Goloborodko et al., 2015*). Simulations of this system at larger scales (*Goloborodko et al., 2015*) have established that there are two regimes of the steady-state dynamics: (i) a sparse regime where little compaction is achieved and, (ii) a dense regime where a dense array of stabilized loops efficiently compacts a chromosome. Importantly, loop extrusion in the dense regime generates chromatin loops that are stabilized by multiple "stacked" condensins (*Alipour and Marko, 2012*), making loops robust against the known dynamic binding-unbinding of individual condensin complexes (*Gerlich et al., 2006*). These two quantitative studies of loop extrusion kinetics (*Goloborodko et al., 2015*; *Alipour and Marko, 2012*) focused on the hierarchy of extruded loops and did not consider the 3D conformation and topology of the chromatin fiber in the formed loop arrays. The question of whether loop-extruding factors can act on a chromatin fiber so as to form an array of loops, driving chromosome compaction and chromatid segregation, as originally hypothesized for condensin in *Nasmyth (2001)*, is salient and unanswered. The main objective of this paper is to test this hypothesis and to understand how formation of extruded loop arrays ultimately leads to compaction, segregation and disentanglement (topology simplification) of originally intertwined sister chromatids.

Here we use large-scale polymer simulations to show that active loop extrusion in presence of topo II is sufficient to reproduce robust lengthwise compaction of chromatin into dense, elongated, prophase chromatids with morphology in quantitative accord with experimental observations. Condensin-driven lengthwise compaction, combined with the strand passing activity of topo II drives disentanglement and segregation of sister chromatids in agreement with the theoretical prediction that linearly compacted chromatids must spontaneously disentangle (*Marko, 2011*).

## Model

We consider a chromosome as a flexible polymer, coarse-graining to monomers of 10 nm diameter, each corresponding to three nucleosomes (600 bp). As earlier (*Naumova et al., 2013*; *Fudenberg et al., 2015*), the polymer has a persistence length of ~5 monomers (3 Kb), is subject to excluded volume interactions and to the activity of loop-extruding condensin molecules and topo II (see below, and in *Fudenberg et al. (2015)*). We simulate chains of 50000 monomers, which corresponds to 30 Mb, close to the size of the smallest human chromosomal arm.

Each condensin complex is modeled as a dynamic bond between a pair of monomers that is changed as a function of time (*Figure 1*, *Video 1*). Upon binding, each condensin forms a bond between two adjacent monomers; subsequently both bond ends of a condensin move along the chromosome in opposite directions, progressively bridging more distant sites and effectively extruding a loop. As in prior lattice models of condensins (*Goloborodko et al., 2015*; *Alipour and Marko, 2012*), when two condensins collide on the chromatin, their translocation is blocked; equivalently, only one condensin is permitted to bind to each monomer, modeling their steric exclusion. Exchange of condensins between chromatin and solution is modeled by allowing each condensin molecule to stochastically unbind from the polymer. To maintain a constant number of bound condensins, when one condensin molecule dissociates, another molecule associates at a randomly chosen location, including chromatin within loops extruded by other condensins. This can potentially lead to formation of reinforced loops (*Video 1*) (*Goloborodko et al., 2015*; *Alipour and Marko, 2012*).

To simulate the strand-passing activity of topo II enzymes, we permit crossings of chromosomal fiber by setting a finite energy cost of fiber overlaps. By adjusting the overlap energy cost we can control the rate at which topology changes occur.

This model has seven parameters. Three describe the properties of the chromatin fiber (linear density, fiber diameter, and persistence length); two control condensin kinetics (the mean linear separation between condensins, and their processivity, i.e. the average size of a loop formed by an isolated condensin before it dissociates, and defined as two times the velocity of bond translocation times the mean residence time); one parameter controls the length of the monomer-monomer bond

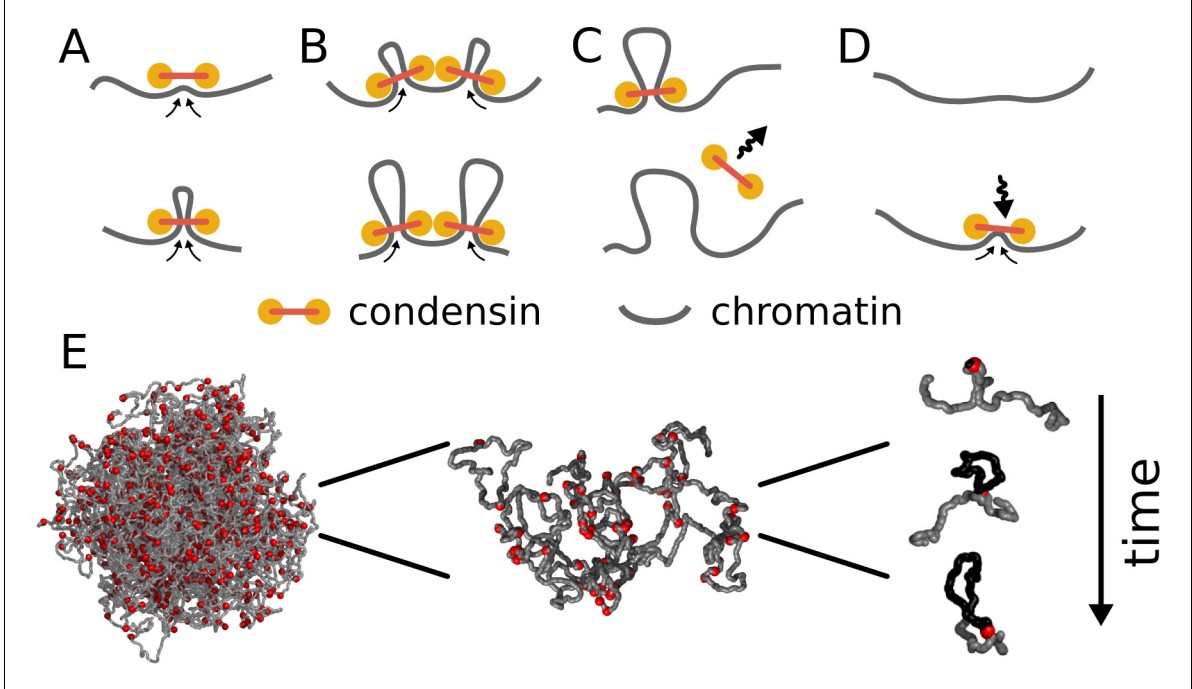

**Figure 1.** Model of loop extrusion by condensins. Top row, the update rules used in simulations: (**A**) a condensin extrudes a loop by moving the two ends along the chromosome in the opposite directions, (**B**) collision of condensins bound to chromosomes blocks loop extrusion on the collided sides, (**C**) a condensin spontaneously dissociates and the loop disassembles; (**D**) a condensin associates at randomly chosen site and starts extruding a loop. Bottom row, (**E**) we use polymer simulations to study how combined action of many loop extruding condensins changes the conformation of a long chromosome.

mediated by a condensin (effectively the size of a condensin complex), and finally we have the energy barrier to topological change.

The effects of two of these parameters, linear separation of condensins and their processivity, have been considered earlier in the context of a 1D model (*Goloborodko et al., 2015*). These parameters control the fraction of a chromosome extruded into loops and the average loop length. In our simulations, we use the separation of 30 kb (1000 condensins per 30 Mb chromosome, as measured in *Fukui and Uchiyama (2007)* and the processivity of 830 kb such that condensins form a dense array of gapless loops with the average loop length of 100 kb, which agrees with available experimental observations (*Paulson and Laemmli, 1977*; *Earnshaw and Laemmli, 1983*; *Naumova et al., 2013*; *Jackson et al., 1990*). The effects of the other five parameters are considered below.

Initially, chromosomes are compacted to the density of chromatin inside the human nucleus and topologically equilibrated, thus assuming an equilibrium globular conformation, corresponding to a chromosomal territory. In the simulations of sister chromatid segregation, the initial conformation of one chromatid is generated as above, while the second chromatid traces the same path at a distance of 50 nm and winds one full turn per 100 monomers (60 kb). In this state, two chromatids run parallel to each other and are highly entangled, while residing in the same chromosome territory. This represents to the most challenging case for segregation of sister chromatids.

## Results

### Loop extrusion compacts randomly coiled interphase chromosomes into prophase-like chromosomes

Our main qualitative result is that loop extrusion by condensins and strand passing mediated by topo II convert initially globular interphase chromosomes into elongated, unknotted and relatively

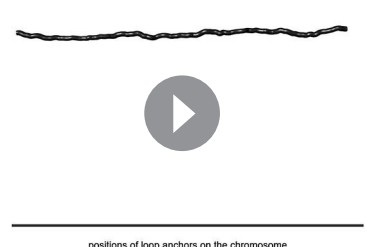

positions of loop anchors on the chromosome

**Video 1.** An illustration of the model of loop extrusion by condensins. When a condensin (red sphere) binds to chromatin (grey fiber), it binds to two adjacent loci and then slides the two contact points in the opposite directions, effectively extruding a loop (highlighted in orange). The diagram on the bottom shows the position of the two loci brought together by the condensin over time (red arc). Condensins occasionally unbind and then rebind to a randomly chosen site, starting a new loop. When two condensins (loops shown in orange and purple) meet on the chromosome, they stop extrusion on the collided sides. Condensins can also bind within already extruded loops. If the growth of the outer loop is blocked by other loops or barriers (not shown), the inner condensin can re-extrude it, forming a single loop reinforced by multiple condensins (the number above the arc). Such a loop is stabilized against unbinding of individual condensins and disassembles only upon unbinding of the last condensin. Available at https://www.youtube.com/watch?v=xikxZ0yF1yU&list=PLnV-JMzdy0ZMOvyK6Z1bzU7C2Jt8Brbit&index=1

stiff structures that closely resemble prophase chromosomes (*Figure 2*, *Video 2*, *Figure 2—figure supplement 1*). The compacted chromosomes have the familiar 'noodle' morphology (*Figure 2BC*) of prophase chromosomes observed in optical and electron microscopy (*Yunis, 1981*; *Sumner, 1991*). Consistent with the observed linear organization of prophase chromosomes, our simulated chromosomes preserve the underlying linear order of the genome (*Yunis, 1981*; *Sumner, 1991*; *Yunis and Sanchez, 1975*; *Furey and Haussler, 2003*; *Strukov and Belmont, 2009*) (*Figure 2C*). Moreover, the geometric parameters of our simulated chromosomes match the experimentally measured parameters of mid-prophase chromosomes *Figure 2—figure supplement 1*: both simulated and human chromosomes have the linear compaction density of ~5 kb/nm (*Yunis, 1981*) and the radius of 300 nm (*el-Alfy and Leblond, 1989*; *Sarkar et al., 2002*; *Kireeva et al., 2004*).

The internal structure of compacted chromosomes also agrees with structural data for mitotic chromosomes. First, loop extrusion compacts a chromosome into a chain of consecutive loops (*Goloborodko et al., 2015*) which agrees with a wealth of microscopy observations (*Paulson and Laemmli, 1977*; *Earnshaw and Laemmli, 1983*; *Maeshima et al., 2005*) and the recent Hi-C study (*Naumova et al., 2013*). Second, our polymer simulations show that a chain of loops naturally assumes a cylindrical shape, with loops forming the periphery of the cylinder and loop-extruding condensins at loop bases forming the core (*Figure 2C*, *Video 2*), as was predicted in *Nasmyth (2001)*. Condensin-staining experiments reveal similar cores in the middle of in vivo and reconstituted mitotic chromosomes (*Maeshima et al., 2005*; *Maeshima and Laemmli, 2003*). In agreement with experiments where condensin cores appear as early as late prophase (*Kireeva et al., 2004*), we observe rapid formation of condensin core in simulations (*Figure 2*, *Video 2*). Thus, the structure of chromosomes compacted by loop extrusion is consistent with the loop-compaction picture of mitotic (*Paulson and Laemmli, 1977*; *Maeshima and Laemmli, 2003*; *Marsden and Laemmli, 1979*) and meiotic chromosomes (*Liang et al., 2015*; *Kleckner, 2006*). However, it should be noted that our model does not rely on a connected 'scaffold', and that cleavage of the DNA will result in disintegration of the entire structure, as is observed experimentally (*Poirier and Marko, 2002*).

## Loop-extrusion generates chromosome morphology kinetics similar to those observed experimentally

Simulations also reproduce several aspects of compaction kinetics observed experimentally. First, the model shows that formed loops are stably maintained by condensins despite their constant exchange with the nucleoplasm (*Hirano and Mitchison, 1994*; *Gerlich et al., 2006*; *Oliveira et al., 2007*). This stability is achieved by accumulation of several condensins at a loop base (*Goloborodko et al., 2015*; *Alipour and Marko, 2012*). Supported by multiple condensins, each loop persists for times much longer than the residency time for individual condensins (*Goloborodko et al., 2015*).

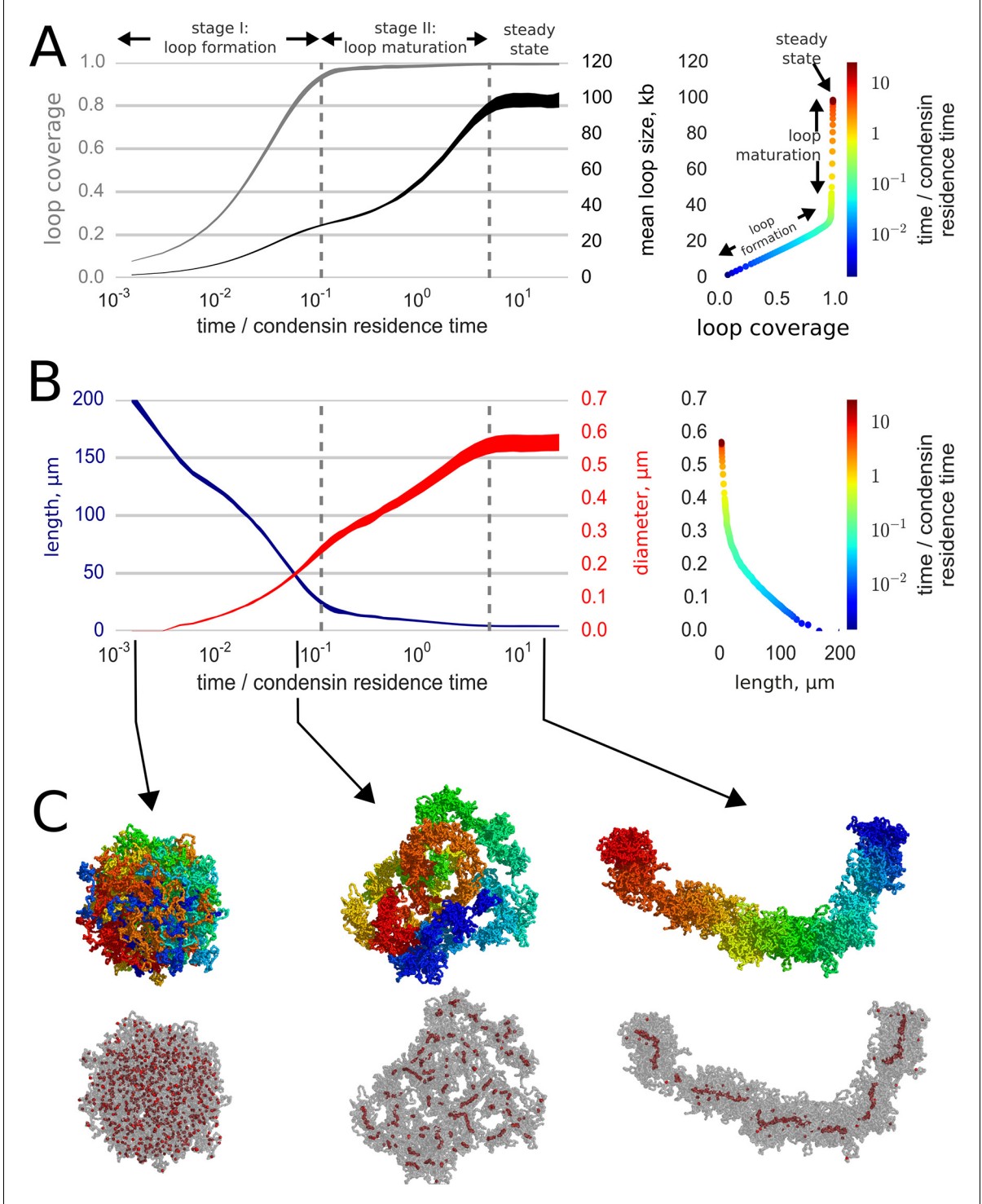

**Figure 2.** Loop-extruding condensins convert globular interphase chromosomes into elongated structures. (**A**) Dynamics of loop array formation. Left, the fraction of the chromosome extruded into loops (loop coverage, gray) and the mean loop size (black). The time is measured in the units of condensin residence time. Data for each curve was obtained using 10 simulation replicas; the thickness of each line shows the difference between the 10th and 90th percentile at each time point. Right, a phase diagram showing loop coverage vs mean loop length. The curve is averaged over the simulation replicas and colored by log-time. Two stages of dynamics are separated by the two vertical dashed lines corresponding to 95% coverage and 95% of the steady state mean loop length. The distinction between the two stages of compaction is especially easy to see on the phase curve. (**B**) Dynamics of chromosome morphology. Left: chromosome length (blue) and the mean chromosome diameter (red) shown as a function of time; the thickness of each line shows the difference between the 10th and 90th percentile at each time point. Right, a phase diagram showing chromosome

*Figure 2 continued on next page*

*Figure 2 continued*

length vs diameter; the curve is averaged over the simulation replicas and colored by log-time. Note that most of chromosome linear compaction is achieved during the first phase, while the second is characterized by widening of the chromosome and further three-fold shortening. (C) Snapshots of polymer simulations. Top row: the conformations of chromosomes at the beginning of simulations (left), slightly before the transition between the loop formation and maturation stages (middle), and in the steady state (right). Color (rainbow: from red to blue) shows positions of individual monomers along the chain and demonstrates that loci in compacted chromosomes are arranged linearly according to their genomic location. Bottom row: same as above, with chromosomes shown in semitransparent gray and condensins as red spheres.

The following figure supplement is available for figure 2:

**Figure supplement 1.** The geometrical parameters of compacted chromosomes in simulations with altered parameters.

Second, simulations show that chromosome compaction proceeds by two stages (*Figure 2AB*): fast formation of a loose, elongated morphology, followed by a slow maturation stage. During the initial 'loop formation stage', condensins associate with the chromosome and extrude loops until colliding with each other, rapidly compacting the chromosome (*Figure 2*, *Video 2*). At the end of this stage, that takes a fraction of the condensin exchange time, a gapless array of loops is formed, the loops are relatively small, supported by single condensins, and the chromosome is long and thin (*Figure 2B*).

During the following 'maturation stage', due to condensin exchange some loops dissolve and divide while others grow and get reinforced by multiple condensins (*Goloborodko et al., 2015*; *Alipour and Marko, 2012*). As we recently showed (*Goloborodko et al., 2015*), a loop can dissolve when all of its condensins dissociate; while new loops are born when two condensins land inside an existing loop at about the same time and split it into two loops. As the mean loop size grows during the maturation stage (*Figure 2A*), the chromosome becomes thicker and shorter. By the end this stage, which takes several rounds of condensin exchange, the rates of loop death becomes equal the rate of loop division and the chromosome achieves a steady state. At steady state, loops get reinforced by multiple condensins and the average loop size and the lengthwise compaction reach their maximal values (*Figure 2*). These dynamics are in accord with the observation that chromosomes rapidly attain a condensed 'noodle'-like shape in early prophase, then spend the rest of prophase growing thicker and shorter (*Yunis, 1981*; *Sumner, 1991*; *Kireeva et al., 2004*; *Schwarzacher, 1976*) (see Discussion).

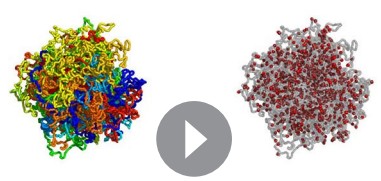

**Video 2.** A representative polymer simulation of compaction of 30 Mb chromosome by loop extruding condensins in presence of strand-passing topo II. Top left, fiber color corresponds to the position of each monomer along the genome. Top right, fiber shown in semitransparent gray, condensins are shown as red spheres. Bottom, the diagram of the crosslinks (red arcs) formed by condensins. The numbers above the arcs show the number of condensins stacked at the bases of reinforced loops. We sped up the second half of the video 20 times to illustrate the slow phase of loop maturation and the long-term stability of the compacted state. Available at https://www.youtube.com/watch?v=_Vc7__xfnfc&list=PLnV-JMzdy0ZMOvyK6Z1bzU7C2Jt8Brbit&index=2

## Loop extrusion separates and disentangles sister chromatids

In the second set of simulations, we studied whether loop extrusion and strand passing simultaneously lead to spatial segregation (*Nasmyth, 2001*) and disentanglement of sister chromatids (*Marko, 2009*). We simulated two long polymers connected at their midpoints that represented sister chromatids with stable centromeric cohesion. To model cohesion of sister chromatids in late G2 phase we further twisted them around each other (see Materials and methods) while maintaining them with a chromosomal territory.

We find that the activity of loop-extruding condensins in the presence of topo II leads to compactions of each of the sister chromatids into prophase-like 'noodle' structures (*Video 3*). Moreover, we observe that formed compact

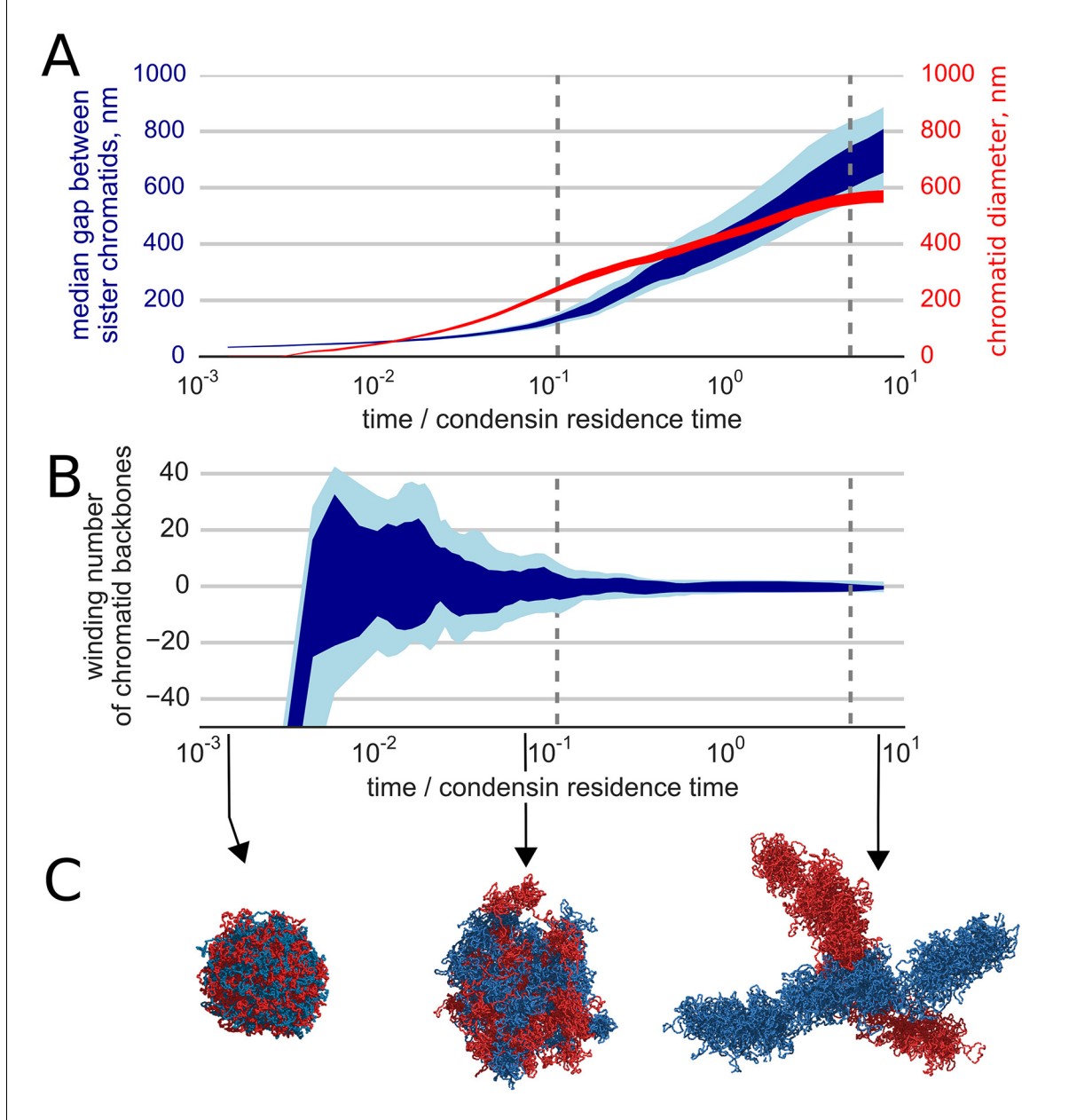

**Figure 3.** Segregation and disentanglement of sister chromatids. (**A**) The median separation between condensin-rich axes of two sister chromosomes (blue) and the mean chromosome diameter (red) as a function of time, measured in condensin residence times, as above. The blue band shows the range between 25th and 75th percentile across 10 simulation replicas at each time point; the light blue shows 10%–90% range. (**B**) Entanglement of sister chromatids measured by the linking number of the condensin cores of sister chromatids; the blue bands show variability between simulation replicas, same as above. (**C**) Conformations of two sister chromatids in the polymer simulations (shown in red and blue) illustrate the observed segregation and disentanglement.

The following figure supplement is available for figure 3:

**Figure supplement 1.** Segregation and disentanglement of sister chromatids in simulations with reduced Topo II concentration.

chromosomes (a) become segregated from each other, as evident by the drastic growth of the mean distance between their backbones (*Figure 3A*), and (b) become disentangled, as shown by the reduction of the winding number of their backbones (*Figure 3B*). Thus, loop extrusion and strand

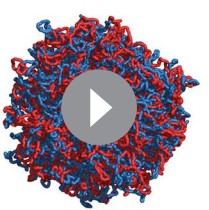



**Video 3.** A polymer simulation of segregation and disentanglement of two sister chromatids by loop extruding condensins in presence of strand-passing topo II. The two sister chromatids (30 Mb each) are shown in red and blue. Initially, the two chromatids were compacted and twisted around each other. Second half of the video is sped up 20x. Available at https://www.youtube.com/watch?v=stZR5s9n32s&index=3&list=PLnV-JMzdy0ZMOvyK6Z1bzU7C2Jt8Brbit

**Video 4.** Sixteen simulation replicas of chromatid segregation and disentanglement. Available at https://www.youtube.com/watch?v=UXXLnGGit-8&index=4&list=PLnV-JMzdy0ZMOvyK6Z1bzU7C2Jt8Brbit

passing fold highly catenated sister chromatids into separate disentangled structures. (*Figure 3C*, *Video 3*).

Despite stochastic dynamics of condensins, compaction, segregation and disentanglement of sister chromatids are highly reproducible and robust to changes in simulation parameters. Disentanglement and segregation were observed in each of 16 simulation replicas (*Video 4*). We also performed simulations with altered chromatin polymer parameters (*Video 5*). Again, we observed chromatids segregation and disentanglement irrespective of fiber parameters and size of a single condensin molecule.

Our model makes several predictions that can be tested against available experimental data. First, in agreement with the published literature (*Savvidou et al., 2005*; *Steffensen et al., 2001*;

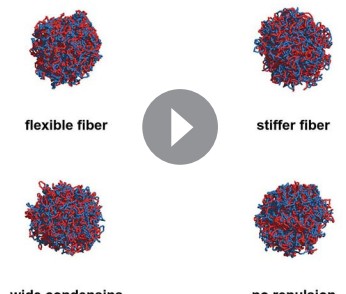

**Video 5.** Polymer simulation of segregation and disentanglement of two sister chromatids with altered model parameters. Top row: left, simulations with a more flexible fiber (0.5x bending energy); right – a stiffer fiber (2x bending energy); bottom row, left - simulations with wider condensins at loop bases; right – simulations with disabled excluded volume interactions. Available at https://www.youtube.com/watch?v=YhgSHppJXVI&index=5&list=PLnV-JMzdy0ZMOvyK6Z1bzU7C2Jt8Brbit

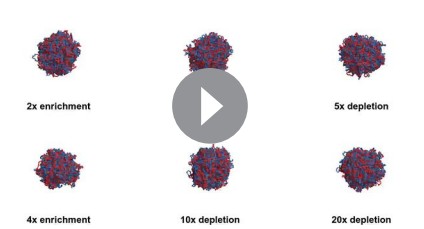

**Video 6.** Polymer simulation of segregation and disentanglement of two sister chromatids with modified condensin abundance. Top row, left - 2000 condensins per chromatid, middle - 500 condensins per chromatid; right - 200 condensins; bottom row, left - 4000 condensins, left - 100 condensins; right, 50 condensins. Available at https://www.youtube.com/watch?v=4img9nmQrtU&index=6&list=PLnV-JMzdy0ZMOvyK6Z1bzU7C2Jt8Brbit

**Video 7.** A polymer simulation of segregation and disentanglement of two sister chromatids upon topo II depletion. Depletion of topo II was simulated by a simultaneous increase of the energy barrier of fiber overlap and the increased radius of repulsion, which decreased, but not fully prevented chain crossings. Available at https://www.youtube.com/watch?v= YO3Qx8liK3s&list=PLnV-JMzdy0ZMOvyK6Z1bzU7C2Jt8Brbit&index=7

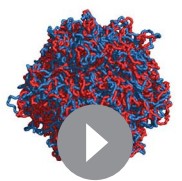

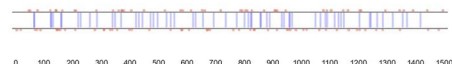

**Video 8.** A polymer simulation of segregation and disentanglement of sister chromatids in the presence of cohesins connecting the chromatids. We simulated cohesins as crosslinks between the chromosomes (shown with the blue lines on the bottom arc diagram) that can be pushed along each of the chromatids by loop-extruding condensins. Available at https://www.youtube.com/watch?v=vmwNbz41pqM&list=PLnV-JMzdy0ZMOvyK6Z1bzU7C2Jt8Brbit&index=8

*Hagstrom et al., 2002*; *Ono et al., 2013*) the abundance of condensins on the chromosome significantly affects geometry of simulated chromatids (*Video 6*). When we varied abundance of chromosome-bound condensin by 20-fold (five-fold decrease and four-fold increase) we observed two major effects (*Figure 2—figure supplement 1*). First, we noticed that condensin depletion leads to a systematic (three-fold) increase in chromosome diameter. Second, we observed nonmonotonic changes in the linear density of chromatin in compacted chromosomes: both five-fold increase and four-fold decrease of condensin abundance lead to about two-fold drop in linear compaction (i.e. two-fold increase in chromosome length). This reduced compaction, however, emerges due to different mechanisms: an increase of condensin abundance leads to reduction of loop sizes while keeping all chromatin in the loops, creating thin chromosomes with long scaffold formed by excessive condensins. Decrease of condensins abundance, on the contrary, leads to formation of chromosomes where some regions are not extruded into loops; such gaps between loops lead to substantial increase in chromosome length and inefficient compaction. These nontrivial dependences of chromosome geometry on condensin abundance can be further tested experimentally.

Next, we examined the role of topological constraints and topo II enzyme in the process of chromosome compaction and segregation. Activity of topo II is modeled by a finite energy barrier for monomer overlap, allowing for occasional fiber crossings. We simulated topo II depletion by elevating the energy barrier and increasing the radius of repulsion between distant particles, thus reducing the rate of fiber crossings. In these simulations, segregation of sister chromatids is drastically slowed down (*Video 7*). This effect of topological constraints on kinetics of compaction and segregation is expected since each chromosome is a chain with open ends that thus can eventually segregate from each other. Reduction of topo II activity (increase of the crossing barrier) prevents disentanglement and segregation of sister chromatids, while having little effect of diameter of individual chromatids (*Figure 3—figure supplement 1*). We conclude that topo II is essential for rapid chromosome segregation and disentanglement, in accord with an earlier theoretical work (*Sikorav and Jannink, 1994*) and with experimental observations (*Wood and Earnshaw, 1990*; *DiNardo et al., 1984*; *Newport and Spann, 1987*; *Shamu and Murray, 1992*).

Finally, we examined the role of cohesin-dependent cohesion of chromosomal arms of sister chromatids. It is known that cohesins are actively removed from chromosomal arms in prophase. Interference with processes of cohesins removal or cleavage leads to compaction into elongated chromosomes that fail to segregate (*Kueng et al., 2006*; *Gandhi et al., 2006*). We simulated prophase compaction in the presence of cohesin rings (*Nasmyth and Haering, 2009*) that stay on sister chromatids and can be pushed around by loop-extruding condensins. If cohesins remain uncleaved on sister chromatids (*Video 8*), we observe compaction of sister chromatids, that remain unsegregated, in agreement with experiments (*Kueng et al., 2006*; *Gandhi et al., 2006*).

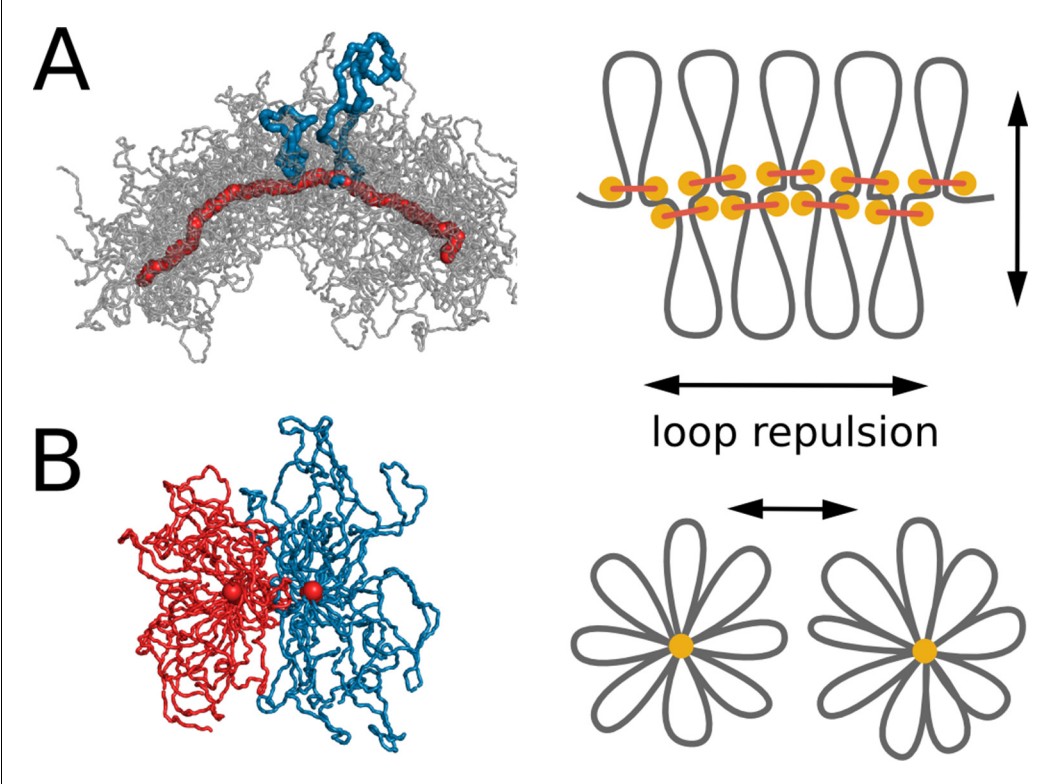

**Figure 4.** Steric repulsion between chromatin loops shapes the compacted chromosomes and leads to segregation of sister chromatids. (A) Left, conformation of a single chromosome with chromatin fiber shown in semitransparent gray and condensins shown in red; two loops are highlighted with blue. Right: the diagram showing that loop repulsion straightens chromosomal cores and extends individual loops radially. (B) Left, cross section of two parallel loop brushes placed at a short distance from each other. Right, the diagram shows that steric interactions between loops of sister chromatids lead to chromatid repulsion and segregation.

Taken together, these results indicate that loop extrusion is sufficient to compact, disentangle and segregated chromosomes from a more open interphase state, to a compacted, elongated and linearly organized prophase state. Future Hi-C experiments focused on prophase chromosomes may be able to make detailed structural comparison with the prediction of our model.

## Repulsion between loops shapes and segregates compacted chromatids

Reorganization of chromosomes by loop extrusion can be rationalized by considering physical interactions between extruded loops. A dense array of extruded loops makes a chromosome acquire a bottle-brush organization with loops forming side chains of the brush. Polymer bottle-brushes has been studied in physics both experimentally and computationally (*Birshtein et al., 1987*; *Ball et al., 1991*; *Li and Witten, 1994*), and has been proposed as models of mitotic and meiotic chromosomes (*Marko and Siggia, 1997*). A recent study also suggested a bottle-brush organization of centromeres in yeast, where centromere chromatin loops lead to separation of sister centromeres and create spring-like properties of the centromere in mitosis (*Lawrimore et al., 2015*; *2016*).

Central to properties of chromosomal bottle-brushes are excluded volume interactions and maximization of conformational entropy that lead to repulsion of loops from each other (*Marko and Siggia, 1997*; *Zhang and Heermann, 2011*). Loops further stretch from the core and extend radially (*Figure 4*), similar to stretching of side chains in polymer bottle-brushes (*Birshtein et al., 1987*; *Ball et al., 1991*; *Li and Witten, 1994*). This in turn leads to formation of the central core where loop-bases accumulate (*Figure 4A*). Strong steric repulsion between the loops, caused by excluded volume interactions, stretches the core, thus maintaining linear ordering of loop bases along the

genome (*Figure 2C*). Repulsion between loops also leads to stiffening of the formed loop-bottle-brush, as chromosome bending would increase overlaps between the loops.

Steric repulsion between loops of sister chromatid leads to segregation of sister chromatids that minimizes the overlap of their brush coronas (*Figure 4B*). Stiffening of the bottle-brush, in turn, leads to disentanglement of sister chromatids. To test this mechanism we performed simulations where excluded volume interactions have been turned off (*Video 5*). In these simulations, chromosomes did not assume a 'bottle-brush' morphology and did not segregate, which confirms the fundamental role of steric repulsion for chromosome segregation.

## Discussion

Our results show that chromosomal compaction by loop extrusion generates the major features of chromosome reorganization observed during prophase (*Nasmyth, 2001*; *Alipour and Marko, 2012*). Loop extrusion leads to lengthwise *compaction* and formation of cylindrical, brush-like chromatids, reducing chromosome length by roughly 25-fold, while at the same time removing entanglements between separate chromatids. Unlike models of prophase *condensation* based on crosslinking of randomly colliding fibers (*Zhang and Heermann, 2011*; *Cheng et al., 2015*), chromatid compaction by loop extrusion exclusively forms bridges between loci of the same chromosome (*Figure 1A*). Similarly, loop-extrusion in bacteria, mediated by SMC proteins, was suggested to form intra-arm chromosomal bridges leading to lengthwise chromosome compaction (*Wang et al., 2015*; *Wilhelm et al., 2015*).

Our computations indicate that it takes an extended period of time for loop-extruding enzymes to maximally compact chromosomes. Maximal compaction is achieved by slow maturation of the loop array over multiple rounds of condensin exchange, rather than a single round of loop expansion. Loop length and the number of condensins per loop gradually increase during the maturation stage. Assuming that condensin II in prophase exchanges with the fast rate of condensin I in metaphase (~200 s) (*Gerlich et al., 2006*) and using an average loop size of 100 kb, and an average spacing of condensins of 30 kb, it should take ~5 condensin exchange times, or about 17 min to achieve maximal compaction, which roughly agrees with the duration of prophase in human cells (*Mora-Bermúdez et al., 2007*; *Leblond and El-Alfy, 1998*). This gradual compaction also may explain why chromosomes of cells arrested in metaphase continue shrinking for many hours after the arrest (condensin II metaphase exchange time ~hours (*Gerlich et al., 2006*).

Our model is idealized and does not aim to describe all of aspects of chromosome folding during mitosis. First, our model does not fully capture the specifics of later stages of mitosis. We model only the action of condensin II in prophase (*Hirota et al., 2004*). Simple geometric considerations, Hi-C studies and mechanical perturbation of mitotic chromosomes show that, in metaphase, cells use additional mechanisms of lengthwise (axial) compaction.

Loop extrusion does not require specific interaction sites (e.g. loop anchors or condensin binding sites) and is therefore robust to mutations and chromosome rearrangements. This robustness also explains how a large segment of chromatin inserted into a chromosome of a different species can be reliably compacted by mitotic machinery of the host (*Dietzel and Belmont, 2001*; *Hirano and Mitchison, 1991*). At the same time, this model allows the morphology of mitotic chromosomes to be tuned. Most importantly, the diameter and length of the simulated chromosomes depend on chromatin fiber properties as well as numbers of condensins (*Figure 2—figure supplement 1*); those dependences invite experimental tests.

In summary, we have shown that loop extrusion is a highly robust mechanism to compact, segregate and disentangle chromosomes. In a recent study (*Fudenberg et al., 2015*) we also demonstrated that loop extrusion by another SMC complex, cohesin, could lead to formation of interphase topological association domains (TADs). We suggest that during interphase, cohesins are sparsely bound, extruding 50–70% of DNA into highly dynamic loops, while metaphase condensins are bound more densely and more processive forming a dense array of stable loops. Taken together these studies suggest that loop extrusion can be a universal mechanism of genome folding, to mediate functional interactions during interphase and to compact during mitosis.

# Materials and methods

## Methods

Our model of loop extrusion acting on a polymer fiber consists of (a) a 1D model that governs the dynamics of intra-chromosomal bonds formed by condensins over time and (b) a 3D polymer model of chromosome dynamics subject to the condensins bonds

## The 1D model of loop extrusion

The dynamics of loop extruding condensins on a chromatin fiber is simulated using a 1D lattice simulations as it was described previously for generic loop-extruding factors (*Goloborodko et al., 2015*; *Alipour and Marko, 2012*). In the lattice, each position corresponds to one monomer in the 3D polymer simulation. We model a condensin molecule very generally as having two chromatin binding sites or, 'heads', connected by a linker. Each head of a condensin occupies one lattice position at a time, and no two heads can occupy the same lattice position. To simulate the process of loop extrusion, the positions of the two condensin heads stochastically move away from each other over time (simulated with the Gillespie algorithm *(Gillespie, 1977)*).

We initialize the simulations by placing condensin molecules at random positions along the polymer chain, with both heads in adjacent positions. To simulate the exchange of condensins between chromatin and solution, condensins stochastically dissociate from the chromatin fiber. Every dissociation event is immediately followed by association of another condensin molecule with the chromatin fiber so that the total number of condensins bound to the chromatin stays constant.

This 1D model has four parameters: the size of the lattice, the number of condensins bound to chromatin, the speed of extrusion, and the average residency time of a condensin on the chromatin fiber. We model a 30 Mb chromosome with a lattice of 50000 sites, 600 bp each. The chromosome is bound by 1000 condensins. Without loss of generality, we set the speed of extrusion to be 1 step per unit time and the condensin residency time at 692 units of time, such that the resulting average loop length is equal 167 monomers, or, 100 kb, close to previous observations in vivo.

## 3D simulations of chromosomes

To perform Langevin dynamics polymer simulations we use OpenMM, a high-performance GPU-assisted molecular dynamics (*API Eastman and Pande, 2010*; *Eastman et al., 2013*). To represent chromatin fibers as polymers, we use a sequence of spherical monomers of 1 unit of length in diameter. Here and below all distances are measured in monomer sizes (~3 nucleosomes, ~10 nm), density is measured in particles per cubic unit, and energies are measured in kT. We use the following parameters of the Langevin integrator: particle mass = 1 amu, friction coefficient = 0.01 ps$^{-1}$, time step = 1ps, temperature = 300 K.

Neighboring monomers are connected by harmonic bonds, with a potential $U = 100(r - 1)^2$ (here and below in units of kT). We model polymer stiffness with a three point interaction term, with the potential $U = 5(1 - \cos(\alpha))$, where alpha is the angle between neighboring bonds.

To allow chain passing, which represents activity of topoisomerase II, we use a soft-core potential for interactions between monomers, similar to *Naumova et al. (2013)*, *Le et al. (2013)*. All monomers interact via a repulsive potential

$$U = 2.0\left(-1 + \left(r/\sqrt{(6/7)}\right)^{12} \cdot \left(\left(r/\sqrt{(6/7)}\right)^2 - 1\right) \cdot (823543)/46656\right)$$

This is a fast and efficient potential designed to be constant at 2.0 kT up to r = 0.7–0.8 and then quickly go to zero at r = 1.00.

To connect 1D LEF simulations with 3D polymer simulations, we first run 1D LEF dynamics for a total period of 10 condensin residence times, recording the state of the systems each unit of time. We then assign bonds to the monomers in polymer simulations according to the current position of condensins' heads. The two monomers held by the two heads of each condensin are connected by a harmonic bond with the potential $U = 100(r - 1)^2$. For each position of condensins' heads from the 1D model, we perform 40000 steps of Langevin dynamics. 3D conformations are recorded every 20000 steps.

We allow an overlap of the heads of collided condensins at the loop bases (i.e. two heads of collided condensins could occupy the same monomer instead of two adjacent monomers). We implement this by shifting the positions of the downstream heads of all condensins by 1 monomer downstream. This allows us to achieve the maximally possible compaction of the chromosomal core (one loop per one monomer of the axis).

We generate the initial conformation of a single chromosome as following: a polymer chain was spherically compacted to a density of 0.01 particle per unit length[3], then allowed to equilibrate over 4,000,000 steps of Langevin dynamics, with a gradual increase of repulsion energy to equilibrate both the topology and the distribution of density inside the confining sphere. We generate the initial conformations of two sister chromatids by winding of two polymer chains along the conformation of a single chromosome, with one full turn each 100 monomers.

We simulate topo II depletion by adjusting two factors. First, we increase the energy of monomer overlap to 20 kT. Since this measure alone proved to be inefficient to prevent chain passing, we additionally increase the radius of repulsion up to 3 length units. In order to maintain the contour length of the polymer, we keep the length of a monomer bond at 1 and ignore repulsion between pairs of neighboring monomers, up to 3 monomers distance along the chain.

To study how the parameters of simulations affect the geometry of compacted chromosomes, we alter the following parameters: (a) increase 2x and (b) decrease 0.5x the number of condensins, (c) increase 2x and (d) decrease 0.5x the bending energy, (e) disallow overlaps of condensins at loop bases, thus simulating wide condensins; (f) reduce linear DNA density of the chromatin fiber to 400 bp/10 nm to model compaction of a 10 nm fiber of stacked nucleosomes and (e) increase linear DNA density of the chromatin fiber to 2400 bp/10 nm and increased the fiber thickness to 30 nm to model compaction of a 30 nm fiber.

## Acknowledgements

Work at NU was supported by the NSF through Grants DMR-1206868 and MCB-1022117, and by the NIH through Grants GM105847 and CA193419. Work at MIT was supported by the NIH through Grants GM114190, R01HG003143, and DK107980. We thank Elnaz Alipour, Job Dekker, Nancy Kleckner, the MIT Biophysics students and the members of Mirny Lab: Geoff Fudenberg, Nezar Abdennur, Christopher McFarland and, especially, E M Breville for stimulating and productive discussions.

## Additional information

### Funding

| Funder | Grant reference number | Author |
| --- | --- | --- |
| National Cancer Institute | CA193419 | Anton Goloborodko<br>Maxim V Imakaev<br>John F Marko<br>Leonid Mirny |
| National Institute of Diabetes and Digestive and Kidney Diseases | DK107980 | Anton Goloborodko<br>Maxim V Imakaev<br>John F Marko<br>Leonid Mirny |
| National Institute of General Medical Sciences | GM114190 | Anton Goloborodko<br>Maxim V Imakaev<br>Leonid Mirny |
| National Institutes of Health | R01HG003143 | Anton Goloborodko<br>Maxim V Imakaev<br>Leonid Mirny |
| National Science Foundation | 1504942 | Anton Goloborodko<br>Leonid Mirny |
| National Institute of General Medical Sciences | GM105847 | John F Marko |
| National Science Foundation | DMR-1206868 | John F Marko |

| National Science Foundation | MCB-1022117 | John F Marko |

The funders had no role in study design, data collection and interpretation, or the decision to submit the work for publication.

## Author contributions

AG, Conception and design, Acquisition of data, Analysis and interpretation of data, Drafting or revising the article; MVI, Conception and design, Analysis and interpretation of data; JFM, LM, Conception and design, Analysis and interpretation of data, Drafting or revising the article

## Author ORCIDs

Anton Goloborodko, http://orcid.org/0000-0002-2210-8616

Leonid Mirny, http://orcid.org/0000-0002-0785-5410

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
