## [Decision Letter]

Thank you for submitting your work entitled "Compaction and segregation of sister chromatids via active loop extrusion" for consideration by *eLife*.

Your article has been favorably evaluated by Kevin Struhl (Senior editor) and three reviewers, one of whom, is a member of our Board of Reviewing Editors.

The reviewers have discussed the reviews with one another and the Reviewing Editor has drafted this decision to help you prepare a revised submission.

Summary:

This manuscript proposes and analyzes a mechanism for the compaction and segregation of chromosomes based on the creation and growth of DNA loops formed by condensins. The major conclusion is that these loops have the effect of converting a globular polymer into an elongated unknotted, stiffer structure that resembles prophase chromosomes. Addition of topo II-like crossovers in the simulation results in a faithful recapitulation of disentanglement and segregation of sister chromatids. The manuscript is well-written and the model underlying the simulations provides a conceptually simple picture that makes a great deal of intuitive sense.

Essential revisions:

1) A concern is that the authors do not seem to be aware of a recent report that reaches quite similar conclusions. In Lawrimore et al. (Lawrimore et al., 2015), the authors demonstrate that loops, i.e. side chains relative to a primary axis, have the effect of elongating and stiffening the chain. Please compare Figure 8 (Lawrimore et al., MCB 2016) to Figure 2 and Figure 4 in the current manuscript. In addition, there is compelling experimental evidence that such a loop organization contributes to tension in mitosis (see Figure 6 in Lawrimore et al., JCB 2015). From the physics perspective of the effect of side chains (i.e. loops) on stiffening the axis, there is a wealth of information (e.g. Panyukov et al., JPCB 2009; Panyukov et al., PRL 2009).

While the inclusion of crossovers into their model does provide insight going beyond what previously has been published, the authors should place their condensing-loop results in the context of earlier published papers and carefully explain what aspects of their study are novel.

2) The authors explain how a 2- or 0.5-fold change in condensin levels would impact the kinetics of loop formation and how long these loops are (Figure 2—figure supplement 1 and a wider range in Video 6). It would be interesting to recapitulate in one figure the effect loop-size distribution (and thus compaction and chromatid thickness) on wider ranges of condensin to inform experiments that could titrate condensin expression levels. Addition of graphs showing the dependence of the important observables (loop size, loop coverage, chromatid thickness and length) on condensin concentration (exchange rate) would significantly strengthen the manuscript. The same for the crossover energy (as a proxy for topo II activity) and its effects on segregation and disentanglement (evolution of median gap between chromatids).

---

## [Author Response]

Essential revisions:

1) A concern is that the authors do not seem to be aware of a recent report that reaches quite similar conclusions. In Lawrimore et al. (Lawrimore et al., 2015), the authors demonstrate that loops, i.e. side chains relative to a primary axis, have the effect of elongating and stiffening the chain. Please compare Figure 8 (Lawrimore et al., MCB 2016) to Figure 2 and Figure 4 in the current manuscript. In addition, there is compelling experimental evidence that such a loop organization contributes to tension in mitosis (see Figure 6 in Lawrimore et al., JCB 2015). From the physics perspective of the effect of side chains (i.e. loops) on stiffening the axis, there is a wealth of information (e.g. Panyukov et al., JPCB 2009; Panyukov et al., PRL 2009).

While the inclusion of crossovers into their model does provide insight going beyond what previously has been published, the authors should place their condensing-loop results in the context of earlier published papers and carefully explain what aspects of their study are novel.

We agree that stiffening and stretching of the main chain by sidechain loops is a well-known phenomenon in polymer physics of bottlebrushes and has been proposed to play a central in chromosome stiffening and segregation by Marko and Siggia (MBoC 1997). In our original submission we cited and discussed several original works on physics of polymer brushes and bottlebrushes and the applications to chromosome biology. In the revision we expand the scope of this discussion (see below).

The works of Lawrimore et al. indeed address a related question of centromere organization in yeast, where centromere chromatin loops lead to separation of sister centromeres and create spring-like properties of the centromere in mitosis, consistent with the mechanism suggested earlier (e.g. Marko and Siggia 1997). We discuss studies of Lawrimore et al. in the revised manuscript. These studies, however, do not challenge the novelty of our contribution, which suggests a specific mechanism of loop extrusion that leads to formation of a dense array of loops, that compacts chromatids and lead to their stiffening, segregations and disentanglement.

To address all of these issues we added a new paragraph to the Results section:

“Reorganization of chromosomes by loop extrusion can be rationalized by considering physical interactions between extruded loops. A dense array of extruded loops makes a chromosome acquire a bottle-brush organization with loops forming side chains of the brush. […] A recent study also suggested a bottle-brush organization of centromeres in yeast, where centromere chromatin loops lead to separation of sister centromeres and create spring-like properties of the centromere in mitosis (Lawrimore et al., 2015; Lawrimore et al., 2016).”

2) The authors explain how a 2- or 0.5-fold change in condensin levels would impact the kinetics of loop formation and how long these loops are (Figure 2—figure supplement 1 and a wider range in Video 6). It would be interesting to recapitulate in one figure the effect loop-size distribution (and thus compaction and chromatid thickness) on wider ranges of condensin to inform experiments that could titrate condensin expression levels. Addition of graphs showing the dependence of the important observables (loop size, loop coverage, chromatid thickness and length) on condensin concentration (exchange rate) would significantly strengthen the manuscript.

Loop-size distribution and its dependence of the linear density and processivity of condensins, in the broad range of these parameters, has been systematically studied in another publication from our group that is cited as ref 16 (Goloborodko, A.; Marko, J. F.; Mirny, L. Mitotic chromosome compaction via active loop extrusion. bioRxiv2015DOI: 10.1101/021642 and Biophysical Journal, 2016, in press). Loop-size distributions can be computed more efficiently by a 1D model of loop extrusion considered there, without a need for 3D molecular dynamics simulations presented here. Computing effects of these parameters on chromatid geometry (thickness and linear density), however, required full 3D simulations. We performed these additional simulations spanning 20-fold range of condensin concentration (shown in an updated Video 6) and developed a new panel for Figure 2—figure supplement 1 to present these results. We discuss these findings in a new paragraph of the main text:

“Our model makes several predictions that can be tested against available experimental data. […] These nontrivial dependences of chromosome geometry on condensin abundance can be further tested experimentally.”

A modified Figure 2—figure supplement 1 presents these results.

The same for the crossover energy (as a proxy for topo II activity) and its effects on segregation and disentanglement (evolution of median gap between chromatids).

We performed several additional simulations that examine effect of the crossover energy (as a proxy for topo II activity) on chromatid geometry and segregation. A new Figure 3—figure supplement 1 presents these results and a new sentence in the main text summarizes these findings making specific predictions for experimental tests of the model:

“Reduction of topo II activity (increase of the crossing barrier) prevents disentanglement and segregation of sister chromatids (Figure 3—figure supplement 1), while having little effect of diameter of individual chromatids (Figure 3—figure supplement 1).”